

# An integrated text mining framework for metabolic interaction network reconstruction

Preecha Patumcharoenpol[1,2,*], Narumol Doungpan[3], Asawin Meechai[1,4], Bairong Shen[2], Jonathan H. Chan[1,3] and Wanwipa Vongsangnak[2,5,*]

[1] Systems Biology and Bioinformatics Laboratory, King Mongkut's University of Technology Thonburi, Bangkok, Thailand
[2] Center for Systems Biology, Soochow University, Suzhou, China
[3] School of Information Technology, King Mongkut's University of Technology Thonburi, Bangkok, Thailand
[4] Department of Chemical Engineering, Faculty of Engineering, King Mongkut's University of Technology Thonburi, Bangkok, Thailand
[5] Department of Zoology, Faculty of Science, Kasetsart University, Bangkok, Thailand
[*] These authors contributed equally to this work.

Corresponding authors
Jonathan H. Chan,
jonathan@sit.kmutt.ac.th
Wanwipa Vongsangnak,
wanwipa.v@ku.ac.th,
fsciwpv@ku.ac.th

## ABSTRACT

Text mining (TM) in the field of biology is fast becoming a routine analysis for the extraction and curation of biological entities (e.g., genes, proteins, simple chemicals) as well as their relationships. Due to the wide applicability of TM in situations involving complex relationships, it is valuable to apply TM to the extraction of metabolic interactions (i.e., enzyme and metabolite interactions) through metabolic events. Here we present an integrated TM framework containing two modules for the extraction of metabolic events (Metabolic Event Extraction module—MEE) and for the construction of a metabolic interaction network (Metabolic Interaction Network Reconstruction module—MINR). The proposed integrated TM framework performed well based on standard measures of recall, precision and F-score. Evaluation of the MEE module using the constructed Metabolic Entities (ME) corpus yielded F-scores of 59.15% and 48.59% for the detection of metabolic events for production and consumption, respectively. As for the testing of the entity tagger for Gene and Protein (GP) and metabolite with the test corpus, the obtained F-score was greater than 80% for the Superpathway of leucine, valine, and isoleucine biosynthesis. Mapping of enzyme and metabolite interactions through network reconstruction showed a fair performance for the MINR module on the test corpus with F-score >70%. Finally, an application of our integrated TM framework on a big-scale data (i.e., EcoCyc extraction data) for reconstructing a metabolic interaction network showed reasonable precisions at 69.93%, 70.63% and 46.71% for enzyme, metabolite and enzyme–metabolite interaction, respectively. This study presents the first open-source integrated TM framework for reconstructing a metabolic interaction network. This framework can be a powerful tool that helps biologists to extract metabolic events for further reconstruction of a metabolic interaction network. The ME corpus, test corpus, source code, and virtual machine image with pre-configured software are available at www.sbi.kmutt.ac.th/~preecha/metrecon.

## INTRODUCTION

Biological literature is vast and quickly growing. Text mining (TM) has become a routine analysis tool for rapidly scanning the entire literature with an essential goal to extract the relationships between named biological entities and concepts. Different examples of TM applications to network construction have been reported, such as protein–protein interactions (*Saetre et al., 2010*; *Kabiljo, Clegg & Shepherd, 2009*; *Airola et al., 2008*; *Srihari & Leong, 2013*), gene–gene relationships in co-expression and regulatory networks (*Rodríguez-Penagos et al., 2007*; *Song & Chen, 2009*; *Van Landeghem et al., 2013*), and gene–disease relationships (*Bell et al., 2011*; *Ozgür et al., 2008*). In addition to a wide range of applications, TM is currently adapted for assisting in compiling relationships of biological data from free texts in biological literature and databases (*Hirschman et al., 2012*; *Neves et al., 2013*). In order to face the challenges due to biological complexity, TM tasks have recently advanced from performing simple interaction extraction towards obtaining a better understanding of the semantics behind biological interactions by analyzing associated events. This task is known as event extraction. This development was presented in the form of the BioNLP Shared Task (BioNLP-ST) (*Kim et al., 2011*), which is a biological community-wide effort to advance the development of natural language processing (NLP). Recently, BioNLP-ST'13 (*Kim, Wang & Yasunori, 2013*) focused on complex relationships, especially related to the topic of biomolecular reactions, pathways and regulatory networks (*Van Landeghem & Ginter, 2011*; *McClosky et al., 2012*; *Gerner et al., 2012*; *Bossy, Bessières & Nédellec, 2013*; *Ohta et al., 2013*). Focusing on metabolic relationships, the Pathway Curation (PC) Task—BioNLP-ST'13 presented by *Ohta et al. (2013)* introduced an event extraction task setting to account for metabolic pathways.

Despite the great interest in the use of TM tools for the extraction and annotation of biological entities of genes, proteins, or simple chemicals through the curation of events and pathways, there have been limited studies to date at a biological system scale (e.g., events with interaction network).

This opens a great challenge for integrating state-of-the-art text mining tasks. Considering closely related prior works (*Humphreys, Demetriou & Gaizauskas, 2000*; *Zhang et al., 2009*; *Rzhetsky et al., 2004*; *Kemper et al., 2010*; *Czarnecki et al., 2012*), for example EMPathIE (*Humphreys, Demetriou & Gaizauskas, 2000*), a template-based TM system, was used to extract information about metabolic reactions along with related contextual information (e.g., source organism and pathway name). When evaluated on a corpus, EMPathIE achieved 23% recall and 43% precision (*Humphreys, Demetriou & Gaizauskas, 2000*). Currently, EMPathIE is no longer under active development (*Humphreys, Demetriou & Gaizauskas, 2000*). In addition, PathBinder (*Zhang et al., 2009*), which is based on a statistical method derived from syntactic and semantic properties of the biomolecular interactions, was used to locate evidence of interaction between two molecules. PathBinder achieved F-score of 71% with a predefined dataset. More generic systems may also be used, such as the GeneWays system for extracting, analyzing, visualizing and integrating molecular pathway data (*Rzhetsky et al., 2004*). Nonetheless, GeneWays shows lack of any published evaluation of its performance with metabolic pathway data and it is not

freely available thus far. Moreover, PathText (*Kemper et al., 2010*) is a pathway curating environment which integrates pathway visualizers, TM systems, and annotation tools into one unified environment. However, PathText (*Kemper et al., 2010*) is not openly available. Recently, *Czarnecki et al. (2012)* developed a rule-based approach to reconstruct *Escherichia coli* metabolic pathways from literature cited in EcoCyc database (*Keseler et al., 2013*). Czarnecki et al. achieved recall and precision of 29–70% and 14–41%, respectively for metabolic reaction extraction method on evaluated pathways. Their results suggest the possibility of automating the process of extracting metabolic interactions from free texts.

Despite these efforts, many TM tools remain restricted, such as not being freely available (e.g., GeneWays and Pathtext) or working with merely provided input data (e.g., Pathbinder). Regarding on machine learning (ML) components of TM tools, their primary difference depends on text mining objective and task. These restrictions reduce a tool's ability to integrate a TM framework for automated extracting metabolic interactions from literature. This is a prevalent problem because metabolic studies rely on biological literature. Considering the process of reconstruction of a metabolic interaction network under normal circumstances, biologists depend on the literature and biological databases for annotation and assignment of genes, enzymes, proteins, and metabolites relationships (*Bordbar & Palsson, 2012*; *Feist et al., 2009*; *Heavner et al., 2012*; *Poolman et al., 2009*; *Liu et al., 2013*). Unless an integrated tool for assisting annotation is available, biologists need to perform gene and functional assignments towards metabolic interaction network using manual curation, which can be both a labor intensive and time consuming task (*Andersen, Nielsen & Nielsen, 2008*; *Baumgartner et al., 2007*).

With the current state of TM research in a metabolic context, we developed an integrated TM framework to meet the above mentioned challenges. The objective of this study is to perform integration of various TM tools to develop a framework to extract metabolic events and further use the framework for reconstructing a metabolic interaction network. To achieve this task, we initially constructed a Metabolic Entities (ME) corpus composed of a representative set of the metabolic events (i.e., events with a mechanical description of the metabolic interaction). We thereafter took the constructed corpus for further use in the development of a TM framework. The TM framework contains two developed modules. The first module, Metabolic Event Extraction (MEE) module, is used for extracting metabolic events from the constructed corpus. The second module, Metabolic Interaction Network Reconstruction (MINR) module, is used for reconstructing metabolic interaction networks. For overall evaluation of the integrated TM framework, the predicted entities and pathways were compared to the manually-curated metabolic entities and pathways in the EcoCyc database and the genome-scale metabolic network of *Escherichia coli*. The integrated TM framework generates metabolic interaction networks in forms of a bipartite metabolic graph and of an enzyme–metabolite interaction pair. The results can be visualized using several types of tools for the task of reconstruction of a metabolic interaction network.

## MATERIALS & METHODS

An overview of the proposed integrated TM framework is depicted in Fig. 1. It is divided into six main steps: namely (i) construction of ME corpus; (ii) construction of test corpus;

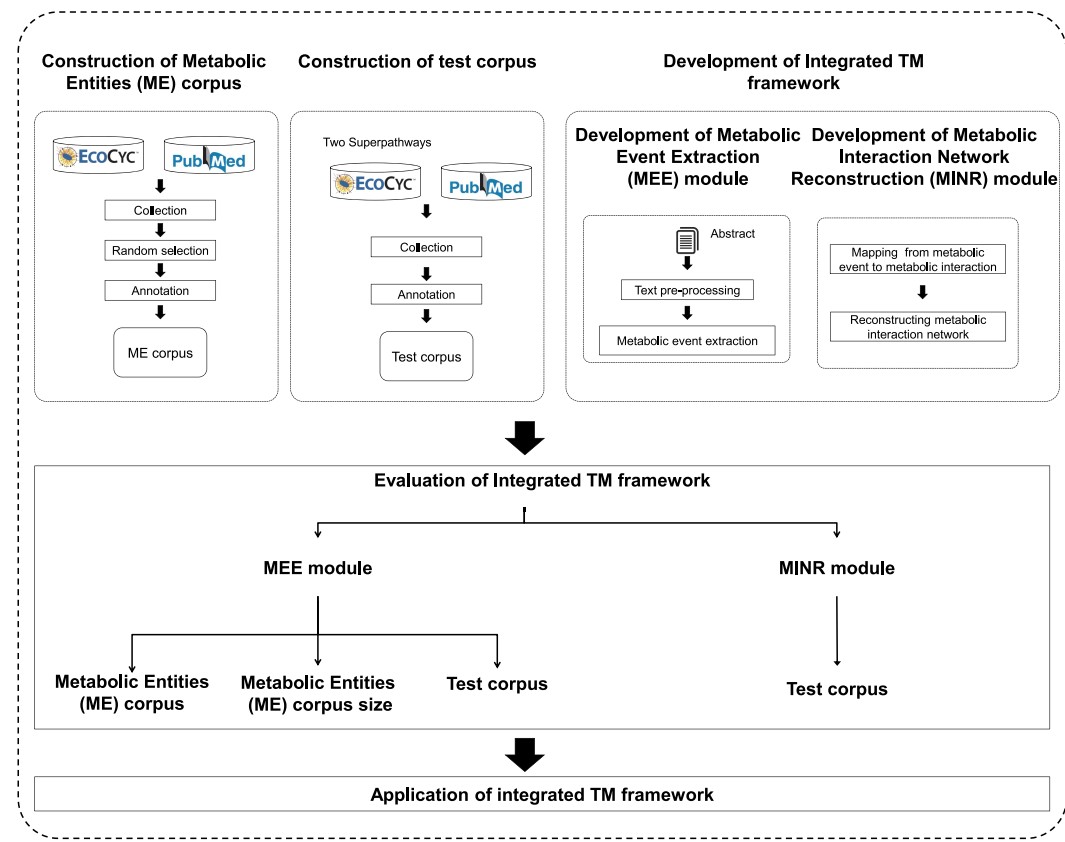

**Figure 1** **A schematic diagram outlining the development steps for the proposed integrated TM framework.** Note: Metabolic entities corpus size means the effect of different corpus sizes on performance of MEE module within the integrated TM framework.

(iii) development of MEE module; (iv) development of MINR module; (v) evaluation of integrated TM framework; and (vi) application of integrated TM framework.

## Construction of Metabolic Entities (ME) corpus

The ME corpus is developed and made publicly available at www.sbi.kmutt.ac.th/~preecha/metrecon. The corpus is licensed under a Creative Commons Attribution-ShareAlike 4.0 International License. It is a collection of various abstracts and titles from different databases that have been manually annotated for metabolic events by two annotators who are biologists with different backgrounds. In order to construct the ME corpus, an article list was initially collected from the EcoCyc database (version 16.5). The EcoCyc database was selected as an example source because it contains a comprehensive resource of biological information for the model organism *E.coli* K12. It contains manually curated and extensive information (e.g., summary comment, regulatory information, literature citation, and extracted evidence type obtained from thousands of publications) (*Keseler et al., 2013*). From this list, articles were randomly selected and their abstracts and titles were then downloaded from the PubMed database for the subsequent annotation process. Considering the process of annotation in each abstract and title, BANNER (*Leaman & Gonzalez, 2008*) was first used to annotate gene and protein (GP), as well as metabolite

**Table 1** Description of entity types.

| Entity type | Reference | Ontology ID |
|---|---|---|
| Gene or Protein (GP) | Ecocyc | SBO:0000246 |
| Metabolite | ChEBI | SBO:0000247 |

**Table 2** Description of metabolic event types.

| Event type | Argument | Description | Ontology ID |
|---|---|---|---|
| Metabolic production | Theme: Metabolite, Cause: Enzyme | Metabolic event that results in formation of metabolite. | SBO:0000176 |
| Metabolic consumption | Theme: Metabolite, Cause: Enzyme | Metabolic event that results in consumption of metabolite. | SBO:0000176 |
| Metabolic reaction | Theme: Metabolite, Cause: Enzyme | Metabolic event that results in conversion of metabolite. | SBO:0000176 |
| Positive regulation | Theme: Event, Cause: Enzyme | Enzyme related to a process that positively regulates a metabolic event. | GO:0048518, GO:0044093 |

entities according to our annotation guideline (www.sbi.kmutt.ac.th/~preecha/metrecon). Table 1 presents annotated entity types along with reference databases i.e., EcoCyc and ChEBI and Systems Biology Ontology (SBO) ID. Afterwards, manual correction using BRAT (*Stenetorp, Pyysalo & Topic, 2012*) by individual domain experts was performed. The metabolic events in the abstracts and titles were annotated according to four types of the metabolic events (i.e., metabolic production, metabolic consumption, metabolic reaction, and positive regulation). Eventually, these annotations were merged to create a final annotation set. For the definition and scope of the metabolic event annotation, the Systems Biology Ontology (SBO) and the Gene Ontology (GO) are considered. Table 2 presents the annotated metabolic event types, arguments and their Ontology ID. A hierarchical representation of the metabolic entities and events is illustrated in Fig. 2A. Also, an example of annotation for metabolic entities and events can be seen in Fig. 2B. For metabolic entities, the annotation identifies phosphoglucosamine mutase and GlmM as GP entities and glucosamine-1-phosphate and glucosamine-6-phosphate as metabolite entities. For metabolic events, the annotation identifies event words of catalyzes and formation as event types of positive regulation and metabolic reaction, respectively. Note that this ME corpus focuses only on metabolic interactions (i.e., enzyme–metabolite interactions) throughout metabolic events at the end. Other types of data, e.g., substrates, products, co-enzymes and co-factors, were considered as metabolites. For discussion of the relation between these entities and event types and the other representations applied in ME corpus, *Ohta et al. (2013)* was used as a reference.

To measure an inter-annotation agreement, 20 abstracts and titles were randomly selected as an example case. The two annotators annotated these abstracts and titles according to the annotation guidelines (www.sbi.kmutt.ac.th/~preecha/metrecon). These annotated abstracts and titles were then used to manually construct consensus annotation.

The annotated results from the two annotators were compared against the constructed consensus annotation and then the F-score was calculated across different data of GP

(A)

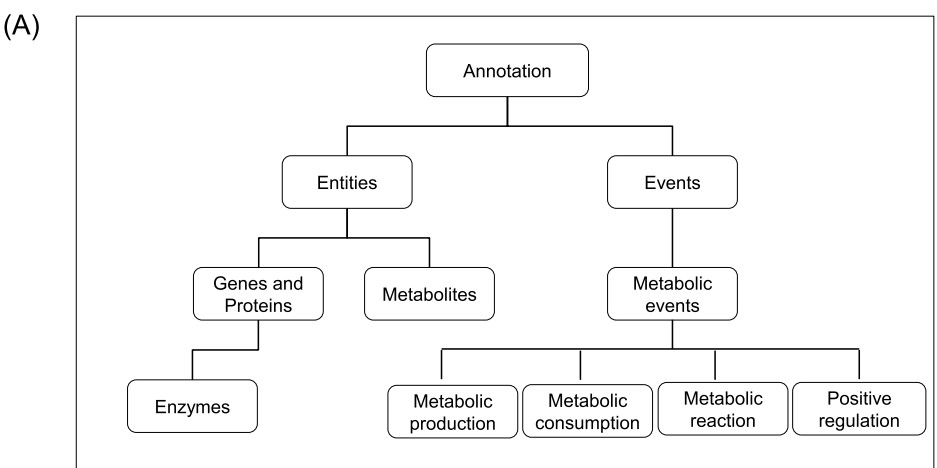

(B)

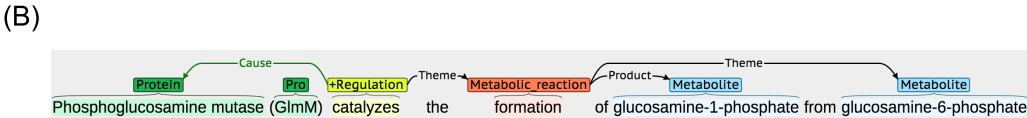

**Figure 2** **A schematic annotation of metabolic entities and events.** (A) Hierarchical representation of metabolic entities and events. (B) An example of metabolic entities and events annotation.

**Table 3** **Inter-annotator agreement of constructed ME corpus.**

| Data | Annotator A | Annotator B | Annotator A/Annotator B |
|---|---|---|---|
| | F-score (%) | F-score (%) | Cohen's kappa coefficient |
| **Entities** | | | |
| GP | 96.17 | 96.03 | 0.96 |
| Metabolite | 93.58 | 91.72 | 0.92 |
| **Events** | | | |
| Metabolic production | 90.24 | 83.33 | 0.72 |
| Metabolic consumption | 96.88 | 85.71 | 0.74 |
| Metabolic reaction | 74.07 | 75.47 | 0.90 |
| Positive regulation | 85.71 | 94.44 | 0.77 |

entities, metabolite entities and events. Additionally, the Cohen's kappa coefficient (*Cohen, 1960*) was also considered as a statistical measure of inter-annotator agreement for GP entities, metabolite entities and events. The overall performance difference between annotators A and B is not significant when compared to consensus annotation (F-scores range from 74.07% to 96.88%) (see Table 3). Also, the agreement between the two annotators were high across all categories (kappa coefficients range from 0.72 to 0.96). It is worth noting that multiple interpretations of numerous entities and events between the two annotators may cause high variability of the inter-annotator agreement level.

## Construction of test corpus

The test corpus (www.sbi.kmutt.ac.th/~preecha/metrecon) was constructed from a collection of introduction, abstract and title sections within two Superpathways articles in

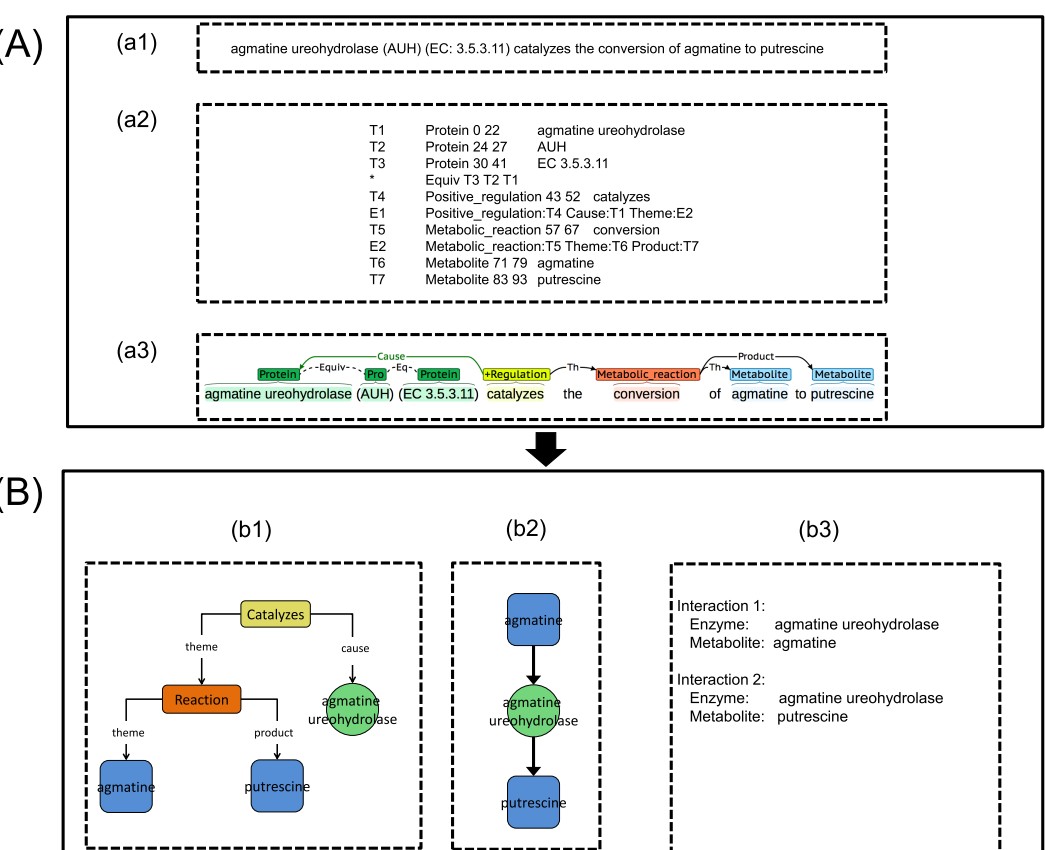

**Figure 3 An illustration showing the process implementation of the proposed integrated TM framework.** (A) Metabolic Event Extraction (MEE) module; (a1) Input text for MEE module, (a2) Output text from MEE module, (a3) Text output visualization using BRAT (*Stenetorp, Pyysalo & Topic, 2012*). (B) Metabolic Interaction Network Reconstruction (MINR) module; (b1) Intermediate graphical output from MINR module, (b2) Graphical output from MINR module, (b3) enzyme–metabolite interactions pair from MINR module.

the EcoCyc and PubMed databases, i.e., the Superpathway of leucine, valine, and isoleucine biosynthesis (18 articles) and the Superpathway of pyridoxal 5′-phosphate biosynthesis and salvage (9 articles). The abstract, introduction and title sections of an article were used because they provide a major summary of the research article, supporting statement and theoretical context. It is also noted that these two selected Superpathways were relatively large compared to others and therefore a considerable number of abstracts, titles and introductions could be collected for test corpus construction. Regarding the annotation process, it was performed as described in the earlier section about the construction of a ME corpus.

## Development of Metabolic Event Extraction (MEE) module

After constructing the ME corpus and test corpus, we developed a MEE module, as illustrated in Fig. 3A. This MEE module architecture was divided into two sub-parts, namely text pre-processing and metabolic event extraction as described below.

### Text pre-processing

The input text (e.g., abstract) as seen in Fig. 3A-a1 was split into sentences using the GENIA Sentence Splitter (*Sætre, Yoshida & Yakushiji, 2007*). Each sentence was then parsed using the McClosky–Charniak–Johnson re-ranking parser (*McClosky, Charniak & Johnson, 2006*). All sentences with more than 100 tokens were skipped and the parser produced a $n$-best list of the $n$ most likely parses of the sentence ($n = 50$ was set as the default parameters in this study). Afterwards, the sentence was converted into collapsed dependency parse using the Stanford parser (*Marneffe, Maccartney & Manning, 2006*). Next, BANNER (*Leaman & Gonzalez, 2008*) was used for detecting all possible GP entities in the sentence. To carry out the subsequent MINR module development work on the metabolic interaction network, we selected only enzyme entities out of all possible GP entities throughout metabolic events by using event word, EcoCyc enzyme name and manual curation. For metabolite entities detection, BANNER trained on the yeast metabolite corpus (*Nobata et al., 2011*) was chosen based on our previous performance evaluation (*Patumcharoenpol et al., 2012*).

### Metabolic event extraction

We retrofitted the Turku Event Extraction System (TEES) by reconfiguration of the entities and the event types as well as retraining TEES in order to support metabolites and metabolic event extraction. Notably, we selected TEES because of its overall performance and ability to handle large-scale datasets (*Gerner et al., 2012*; *Björne, Van Landeghem & Pyysalo, 2012*). TEES is an event classification tool, which utilizes various features from syntax analysis, such as tokens and dependency graphs to identify the probability of metabolic interaction between two entities (*Björne & Salakoski, 2011*). The TEES parameter was estimated from its internal Support Vector Machine while training using a grid search. The retrofitted TEES can be found in the virtual machine image available at www.sbi.kmutt.ac.th/~preecha/metrecon.

## Development of Metabolic Interaction Network Reconstruction (MINR) module

Once MEE module was developed, we further developed a MINR module. We divided the MINR module into two sub-sections: the mapping from metabolic event to metabolic interaction and reconstructing the metabolic interaction network by combinations of individual metabolic interactions.

### Mapping from metabolic event to metabolic interaction

Extracted metabolic events together with enzyme and metabolite entities obtained from the MEE module (Figs. 3A-a2 and 3A-a3) were initially converted into a graph called an event interaction graph in the MINR module (Fig. 3B-b1). This step represented the whole event in a graph format. The event interaction graph was then transformed into a metabolic interaction graph (Fig. 3B-b2). To simplify the events for interaction extraction, the redundancy of the extracted events across articles was eliminated by rearrangement and deletion of nodes based on a predefined rule. File S1 shows the pseudocode used for the MINR module development in converting an event interaction graph to a metabolic

interaction graph, as illustrated in Figs. 3B-b1–3B-b2. The extracted enzyme–metabolite interaction pair could then be eventually obtained, as shown in Fig. 3B-b3

### Reconstructing metabolic interaction network

After mapping from the metabolic event to metabolic interaction, all possible extracted enzyme–metabolite interaction pairs were pooled together. Subsequently, all unique enzyme–metabolite interaction pairs were merged together and then mapped onto a metabolic interaction network. Metabolites were connected through shared enzymes. The resulting metabolic interaction network was enforced to be a bipartite metabolic graph (i.e., enzymes and metabolites). In the case of an incomplete metabolic interaction (e.g., no enzyme for the connecting metabolites), a missing node was filled with a proxy node in order to conform to a bipartite metabolic structure.

## Evaluation of integrated TM framework

To evaluate the integrated TM framework, we assessed the MEE module and the MINR module, separately as described below. MEE module evaluation on three critical sub-parts: ME corpus, ME corpus size and test corpus was performed. In each sub-part, we calculated performance based on standard precision, recall, and F-score as performance measures (*Van Rijsbergen, 1979*).

To compare the obtained results with manually curated entities, we applied the *sloppy span matching* criterion to entities, which means that entities must match the types, but are not required to exactly match the entities boundaries (*Czarnecki et al., 2012*). For metabolic events, the comparison criterion that is termed *approximated boundary matching* (*Kim et al., 1979*) was used. In particular, the three criteria used were: (i) identical metabolic event type, (ii) sloppy span matching between metabolic event trigger span, and (iii) at least one matching or all matching of arguments. We used these criteria in this study due to strong supporting evidence that they are better in terms of information retrieval than the alternative exact matching criterion, where boundaries between two entities are required to match exactly (*Kabiljo, Clegg & Shepherd, 2009*; *Shepherd & Kabiljo, 2008*).

### MEE module evaluation on ME corpus

We performed five-fold cross-validation on the abstracts and titles for error estimation of MEE module. The total number ($D$) of the abstracts and the titles within ME corpus was randomly partitioned into five approximately equal numbers ($D_1, D_2 \ldots, D_5$). An individual fold was iteratively left out and used as the testing dataset while the remaining data were used as the training dataset.

### MEE module evaluation on ME corpus size

In addition to assessing the MEE module using the whole ME corpus constructed in this work, we also evaluated the effect of various corpus sizes. For this part, we performed five-fold cross-validation on subsets of the ME corpus with different abstract sizes. To create these subsets, 100, 150, and 200 abstracts and titles were randomly selected from the constructed ME corpus. To this end, we compared the results with that obtained from the whole corpus of 271 abstracts and titles, which was used to determine an upper bound of F-score.
### MEE module evaluation on test corpus

We evaluated the entities prediction performance of the MEE module using the test corpus as a reference. That is, we ran entities prediction using the MEE module on the test corpus. The predicted results of GP and metabolite entities were then compared to the manually-curated entities in test corpus.

### MINR module evaluation on test corpus

A list of manually-curated metabolic interactions was initially prepared for the MINR module evaluation on the test corpus as provided in File S2. After applying the MINR module on the test corpus for further metabolic interaction network reconstruction, the predicted results were then compared to the manually-curated metabolic interactions list (File S2) for evaluation throughout calculation of the recall, the precision, and the F-score.

## Application of integrated TM framework

To demonstrate the application of the integrated TM framework, we presented two case studies. The first case study showed a comparative analysis of our integrated TM framework with another TM system developed by *Czarnecki et al. (2012)* on the test corpus for reconstruction of the Superpathway of leucine, valine, and isoleucine biosynthesis. To elaborate, we ran our integrated TM framework on the test corpus to extract the metabolic events and mapped them to metabolic interactions for the Superpathway reconstruction. Once completed, we then performed a comparative analysis of the reconstructed network with the other results achieved by the TM system developed by *Czarnecki et al. (2012)*.

For the second case study, the integrated TM framework with EcoCyc extraction was applied to reconstruct a metabolic interaction network. EcoCyc extraction is a collection list of references from the EcoCyc database (2,373 abstracts and titles). In brief, we first ran the integrated TM framework on EcoCyc extraction in order to extract a list of enzyme, metabolite, and enzyme–metabolite interactions association with metabolic events. Also, we ran the integrated TM framework on the ME corpus for performance comparison. To evaluate the performance, the predicted entities in terms of enzymes, metabolites, and enzyme–metabolite interactions were compared to the manually-curated metabolic entities in the EcoCyc database for calculating the precision. A published genome-scale metabolic network of *E. coli* K-12 MG1665 (*i*JO1366) (*Orth et al., 2011*) was also used as the manually-curated metabolic pathways for metabolic interaction network reconstruction.

The integrated TM framework could be run using the virtual machine image on one of Mac, Windows, or Linux system with the pre-configured software available at www.sbi.kmutt.ac.th/~preecha/metrecon. The source code in the virtual machine is licensed under Apache License 2.0.

## RESULTS AND DISCUSSION

The ME corpus and an integrated TM framework for the reconstruction of a metabolic interaction network were developed in this study. The results and discussion are provided below.

**Table 4  Basic statistics of ME corpus and test corpus.**

**Metabolic entities corpus**

| Features | # Units |
|---|---|
| Abstract | 271 |
| Sentence | 2,288 |
| **Entity type** | **Metabolic entities count** |
| Metabolite | 1,898 (7.00[a], 6.52) |
| GP | 2,513 (9.27[a], 8.02) |
| Total | 4,411 (16.28[a], 10.74) |
| **Event type** | |
| Metabolic production | 115 (0.42, 0.99)[a], (1.10[b]) |
| Metabolic consumption | 132 (0.49, 1.07)[a], (1.22[b]) |
| Metabolic reaction | 134 (0.49, 0.94)[a], (2.12[b]) |
| Positive regulation | 99 (0.36, 0.65)[a], (1.76[b]) |
| Total | 480 (1.77, 2.35)[a], (1.51[b]) |

**Test corpus**

| Features | # Units |
|---|---|
| Abstract | 27 |
| Introduction | 24 |
| Sentence | 422 |
| **Entity type** | **Metabolic entities count** |
| Metabolite | 747 (14.09, 12.69)[a] |
| GP | 675 (12.74, 11.55)[a] |
| Total | 1,422 (26.83, 21.73)[a] |
| **Event type** | |
| Metabolic production | 99 (1.87, 2.18)[a] |
| Metabolic consumption | 45 (0.85, 1.52)[a] |
| Metabolic reaction | 62 (1.17, 2.03)[a] |
| Positive regulation | 34 (1.17, 2.03)[a] |
| Total | 240 (4.53, 4.74)[a] |

**Notes.**

[a] The average number of units and per abstract and Standard Deviation (SD).

[b] The average number of arguments per metabolic event.

## ME corpus statistics

Our constructed ME corpus consists of annotated GP, metabolite entities, and metabolic events. Table 4 shows the basic statistics for the constructed ME corpus. Of the 271 abstracts and titles selected from the PubMed database (see 'Methods'), we found a total number of 2,513 entities for GP and 1,898 entities for metabolite, corresponding to 9.27 and 7.00 entities per abstract for GP and metabolite, respectively.

In addition, we further examined the basic statistics on the average number of arguments per metabolic event. As also shown in Table 4, the metabolic reaction event has an average number of 2.12 arguments per metabolic event. This is higher than the average number of arguments in metabolic production event (1.10) and metabolic consumption event
**Table 5 Performance of MEE module on metabolic entities corpus using five-fold cross-validation.**

| Event type | Recall (%) | Precision (%) | F-score (%) |
|---|---|---|---|
| Metabolic production | 62.94 | 55.79 | 59.15 |
| Metabolic consumption | 41.67 | 58.27 | 48.59 |
| Metabolic reaction | 24.50 | 33.56 | 28.32 |
| Positive regulation | 30.70 | 45.57 | 36.69 |
| Micro average | 38.16 | 48.18 | 42.59 |

(1.22). In order to further express the word preference in the metabolic event, we inspected common words in the ME corpus. The top ten list of event words identified in the ME corpus is presented in File S3. Interestingly, we found that a major portion of these common words (45.84% of total metabolic events) are involved in the generic description of biological processes (e.g., Catalyzes, Biosynthesis, Synthesis, Formation, Conversion, Utilization, Catalyzed, Catalyze, and Metabolism). These results showed that most of the event words in the metabolic event were centred around a small set of general keywords. However, it is possible to deduce the types of enzymatic reactions using name of substrate and product in some cases, e.g., the formation of glucosamine-1-phosphate (product name) from glucosamine-6-phosphate (substrate name) (Fig. 2B) suggests a phosphorylation reaction.

## Performance of MEE module on ME corpus

Using five-fold cross-validation on the ME corpus, the recall, the precision, and the F-score were calculated for each metabolic event type as measures for the overall performance evaluation. Table 5 shows these measures for each metabolic event type as well as for the total event type which indicates the sum of all metabolic event types. That is, the measures of total event type were calculated by summation of the individual true positives, false positives, and false negatives for each metabolic event type.

The F-scores of the events of metabolic production (59.15%) and metabolic consumption (48.59%) turned out to be higher than both of the events of metabolic reaction (28.32%) and positive regulation (36.69%). It is intuitive to think that complex events (i.e., two or more arguments) as found in metabolic reaction and positive regulation are harder to be classified than simple events (i.e., one argument) as found in metabolic production and metabolic consumption. In such a case, when the average number of arguments per metabolic event is high, a low F-score is clearly shown (see Tables 4 and 6) as in the example of the metabolic reaction and the positive regulation. These above-mentioned results are strongly supported by earlier works in Pathway Curation (PC) task—BioNLP-ST'13 and BioNLPST'11 (Ohta et al., 2013; Kim et al., 2011). In particular, the F-scores achieved from positive regulation event between PC task—BioNLP-ST'13 (Ohta et al., 2013) and our study were compared. Consequently, the F-scores were similar with values of 39.23 and 36.69, respectively.

## Performance of MEE module on different ME corpus sizes

To assess the effect of ME corpus sizes, different subsets of 100, 150, and 200 abstracts and titles were randomly extracted and compared to the whole ME corpus size of 271 abstracts and titles. Each subset was evaluated three times, and the results shown were averages

**Table 6** Performance of MEE module on test corpus for tagging GP and metabolite.

|  | GP entities | Metabolite entities |
|---|---|---|
| Superpathway of leucine, valine, and isoleucine biosynthesis | | |
| Recall (%) | 81.79 (274/335)[a] | 85.51 (301/352)[a] |
| Precision (%) | 92.88 (274/295)[b] | 91.77 (301/328)[b] |
| F-score (%) | 86.98 | 88.53 |
| Superpathway of pyridoxal 5′-phosphate biosynthesis and salvage | | |
| Recall (%) | 84.62 (297/351)[a] | 65.16 (245/356)[a] |
| Precision (%) | 83.90 (297/354)[b] | 87.81 (245/279)[b] |
| F-score (%) | 84.26 | 74.81 |

**Notes.**

[a] The number in parenthesis indicates correctly predicted entities/number of total correct entities.

[b] The number in parenthesis indicates correctly predicted entities/number of total predicted entities.

Performance of MEE module on test corpus for tagging GP and metabolite used the trained model from five fold cross-validation (see Table 5).

(Fig. 4). The corresponding recall, precision and F-score were compared in the form of learning curves in Fig. 4 using five-fold cross-validation. Expectedly, the performance of MEE module with the largest corpus size was better than that of the smaller corpus sizes in all possible cases. Clearly, the best recall and F-score were obtained with the whole corpus size of 271 abstracts and titles (Figs. 4A and 4C). In general, the trend was improved performance with a larger corpus size for these two measures. A similar trend was observed for the precision measure, except for the case of metabolic production which showed no dependence on corpus size (Fig. 4B). From the overall results, we suggest that a minimum of 150 abstracts and titles should be used for development of a ME corpus. Note that the regularity of the metabolic event description is applied for development of ME corpus for easier event extraction.

## Performance of MEE module on test corpus

As a further assessment of the proposed integrated TM framework, a constructed test corpus was used for performance evaluation of the MEE module. Basic statistics of test corpus can be seen in Table 4. It contained 27 articles related to two Superpathways from the EcoCyc database (see 'Methods'). At first, we evaluated MEE module on the test corpus using GP and metabolite entity tagging. As shown in Table 6, the precision, recall, and F-score of the entity tagger for GP and metabolite were very high for more than 80% of GP and metabolite entities identified for the Superpathway of leucine, valine, and isoleucine biosynthesis. However, the recall and F-score of the entity tagger for metabolite was lower than our expectations for the Superpathway of pyridoxal 5′-phosphate biosynthesis and salvage. The recall showed less than 70%, and the F-score showed less than 80%. These results seem to indicate that the entity tagger (i.e., BANNER) has a weakness in detecting metabolite entities in an abbreviated form (e.g., Pyridoxine (PN), Pyridoxal (PL), and 4-hydroxy-l-threonine phosphate (HTP)).

## Performance of the MINR module on test corpus

We evaluated the performance of the MINR module using the test corpus. In terms of enzyme–metabolite interaction, we found that the MINR module showed high performance

(A)

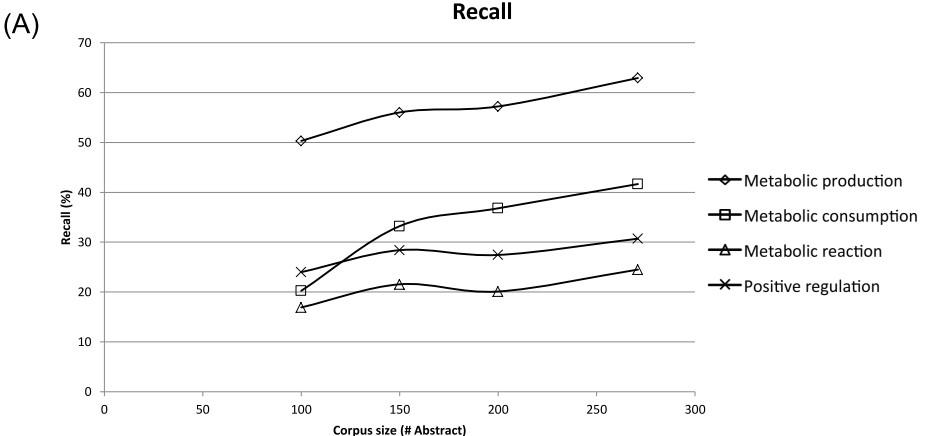

(B)

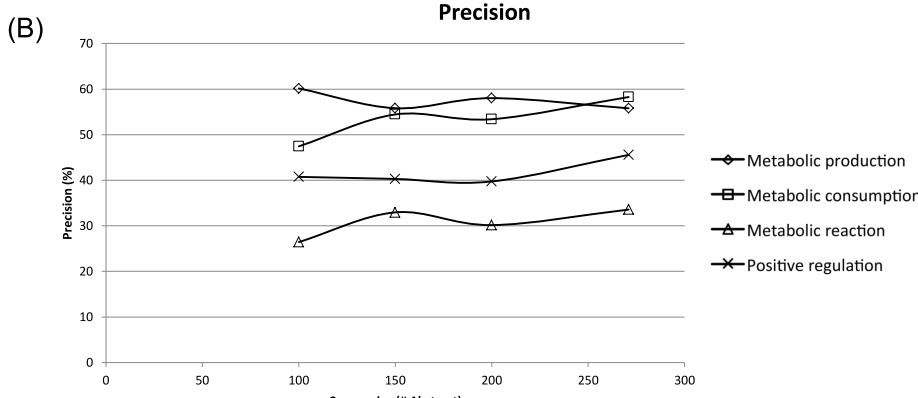

(C)

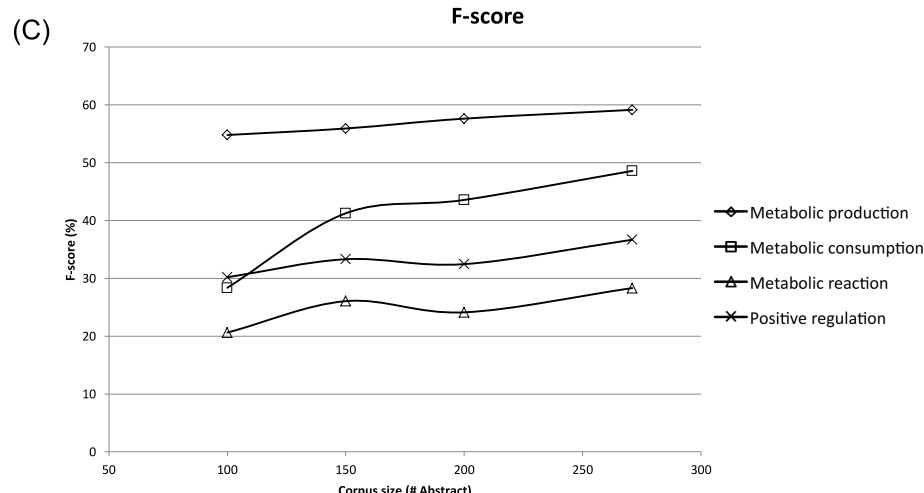

**Figure 4    Learning curves showing performance of the Metabolic Event Extraction (MEE) module on the different corpus sizes using five-fold cross-validation.** Each of metabolic event types and total are plotted against (A) Recall, (B) Precision, and (C) F-score. The total was calculated by summation of the individual true positives, false positives, and false negatives for each of metabolic event types.

**Table 7  Performance of MINR module on test corpus.**

| | Enzyme–metabolite interaction |
|---|---|
| [a]Superpathway of leucine, valine, and isoleucine biosynthesis | |
| Recall (%) | 64.65 (64/99)[c] |
| Precision (%) | 84.21 (64/76)[d] |
| F-score (%) | 73.14 |
| [b]Superpathway of pyridoxal 5′-phosphate biosynthesis and salvage | |
| Recall (%) | 76.84 (73/95)[c] |
| Precision (%) | 90.12 (73/81)[d] |
| F-score (%) | 82.95 |

Notes.

[a]Under the superpathway of leucine, valine, and isoleucine biosynthesis, 88 identified enzyme–metabolite interactions were found by manual curation and used as a reference for performance evaluation.

[b]Under the superpathway of pyridoxal 5′-phosphate biosynthesis and salvage, 87 identified enzyme–metabolite interactions were found by manual curation and used as a reference for performance evaluation.

[c]The number in parenthesis indicates correctly predicted entities/number of total correct entities.

[d]The number in parenthesis indicates correctly predicted entities/number of total predicted entities.

Performance of MINR module on test corpus used the trained model from five fold cross-validation (see Table 5).

as presented in Table 7. The reconstructed results from the MINR module were then compared with manually curated metabolic interactions list (see File S2). As shown in Table 7, for the two Superpathways, the precisions were 80–90%, the F-scores were 70–80%, and the recalls were 60–70%. These results suggest that the MINR module performed well for the mapping of enzyme–metabolite interactions and can be further used for reconstruction of metabolic interaction networks. Nonetheless, there were still missing interactions which could not be identified by the MINR module. These could be because MINR module was unable to capture the metabolic events that were implicit in the text as seen in the example of PMID-13405870 (see File S4). Moreover, it was also unable to extract exact precedence relationships among metabolic events as seen in the example of PMID-13727223 (see File S4). Further improvements are planned for MINR module after taking into account these limitations.

## The integrated TM framework for reconstructed metabolic interaction network

As mentioned in the 'Methods,' two case studies were used to evaluate the integrated TM framework. For the first case study, as shown in Fig. 5, our integrated TM framework on the test corpus successfully extracted 11 entities of enzymes and metabolites as well as 10 enzyme–metabolite interactions for reconstruction of the Superpathway of leucine, valine, and isoleucine biosynthesis.

To elaborate how a biologist can apply our integrated TM framework for reconstruction of the Superpathway of leucine, valine, and isoleucine biosynthesis, we show two example sentences extracted from PMID-1646790 that were obtained from MEE and MINR modules. The examples are described below.

*Example 1: "leucine synthesis by the tyrosine-repressible transaminase in Escherichia coli K-12."*
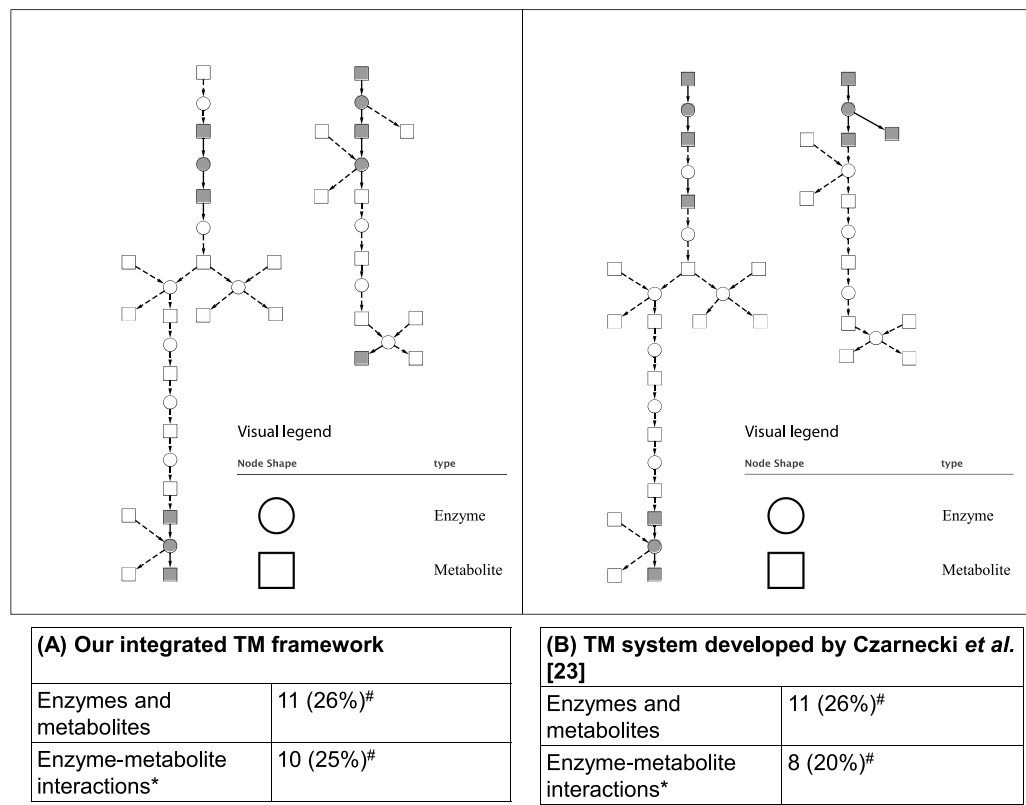

| (A) Our integrated TM framework | |
|---|---|
| Enzymes and metabolites | 11 (26%)[#] |
| Enzyme-metabolite interactions* | 10 (25%)[#] |

| (B) TM system developed by Czarnecki et al. [23] | |
|---|---|
| Enzymes and metabolites | 11 (26%)[#] |
| Enzyme-metabolite interactions* | 8 (20%)[#] |

**Figure 5** **Comparative analysis of two TM systems on test corpus for the Superpathway reconstruction of leucine, valine, and isoleucine biosynthesis.** (A) Our integrated TM framework (MEE and MINR modules), (B) TM system developed by *Czarnecki et al. (2012)*. The extracted interactions were overlaid to the reference metabolic pathway from the EcoCyc database. A grey node means correct extraction. A white node means incorrect extraction. An edge with an arrow means correct extraction. An edge with a dash arrow means incorrect extraction. This bipartite metabolic graph is created using Cytoscape version 3.0.1.
* The extracted enzyme–metabolite interaction indicates the number of binary interaction; #The number in parenthesis is a percentage of extracted mentions/events in comparison to the reference metabolic pathway from the EcoCyc database.

*Example 2: "2-KIC amination by the tyrB-encoded transaminase and also by the aspC- and avtA-encoded transaminases."*

For explanation of example 1, the MEE module identifies enzyme (tyrosine-repressible transaminase) and metabolite (leucine) throughout metabolic production event. Considering example 2, the MEE module identifies enzyme (tyrB-encoded transaminase) and metabolite (2-KIC) throughout metabolic consumption event. To the end, the MINR module obtains enzyme–metabolite interactions by combining the metabolic production and consumption events from examples 1 and 2, respectively. As a result, 2-KIC can be converted to leucine by tyrB-encoded transaminase (tyrosine-repressible transaminase). Full details of enzymes, metabolites, and enzyme–metabolite interactions can be seen in File S5.

Comparing these results to those obtained by a TM system developed by *Czarnecki et al. (2012)*, we found that a similar number of correctly extracted entities of enzymes and

**Table 8** The reconstructed metabolic interaction network using ME corpus and EcoCyc extraction applications.

| Entities | EcoCyc extraction (2,373 abstracts and titles) | | | ME corpus (271 abstracts and titles) | | |
|---|---|---|---|---|---|---|
| | # Correctly predicted | # Total predicted | Precision (%) | # Correctly predicted | # Total predicted | Precision (%) |
| Enzyme | 193 | 293 | 69.93 | 58 | 74 | 78.38 |
| Metabolite | 190 | 260 | 70.63 | 80 | 113 | 70.80 |
| Enzyme–metabolite interaction | 234 | 501 | 46.71 | 76 | 137 | 55.47 |

**Notes.**

ME corpus and EcoCyc extraction was done using the same training dataset (i.e., ME corpus for 271 abstracts and titles). To reconstruct the metabolic interaction network, *E. coli* K-12 MG1665 (*i*JO1366) genome-scale metabolic network (*Orth et al., 2011*) was used as an interaction reference.

metabolites and enzyme–metabolite interactions were obtained (Fig. 5). As can be seen, the results achieved from both systems are able to extract a different part of network suggesting that the combination of the results and biological intepretation would be an interesting option for a biologist who searches for an alternative way for reconstructing a network.

For the second case study involving large-scale data extraction from EcoCyc, the results are shown in Table 8. We found the precisions for this EcoCyc extraction data in terms of enzymes (69.93%), metabolites (70.63%), and enzyme–metabolite interactions (46.71%). After comparing these precisions to the similar results gained from constructed ME corpus (271 abstracts and titles), we found that the EcoCyc extraction data showed a higher number of false positives. Based on our manual inspection, one source of false positives came from mentions of enzymes, metabolites, or enzyme–metabolite interactions in other species that were not from *E. coli* despite the fact that our framework was trained using *E. coli* abstracts and titles. However, this is favorable for the real-world usage since it shows the generality of our method can capture all generic mentioned reactions in text. Another note is that we did not deploy a normalization method in our evaluation, and this might not correctly reflect the performance of the real-world large-scale extraction where the normalization method is critical. Nevertheless, these results illustrate that our constructed ME corpus within the integrated TM framework is solid and can be used as a representative dataset for large-scale data extraction with applications for building metabolic interaction databases and networks as well as for knowledge discovery tasks. The proposed integrated TM framework application is summarized in Fig. 6.

## CONCLUSIONS

This study reports the first open-source integrated TM framework for reconstructing a metabolic interaction network. Here, we constructed a ME corpus, a MEE module and a MINR module within an integrated TM framework. Expectedly, the ME corpus has been successfully used for simplified detection of GP, metabolites entities throughout metabolic events. In addition, we have shown that our proposed framework successfully extracted a metabolic interaction, and it can be used as a scaffold for futher reconstruction of a large-scale metabolic interaction network.

From the overall performance evaluation, the two developed modules within the framework performed well. Using five-fold cross-validation in the MEE module on the
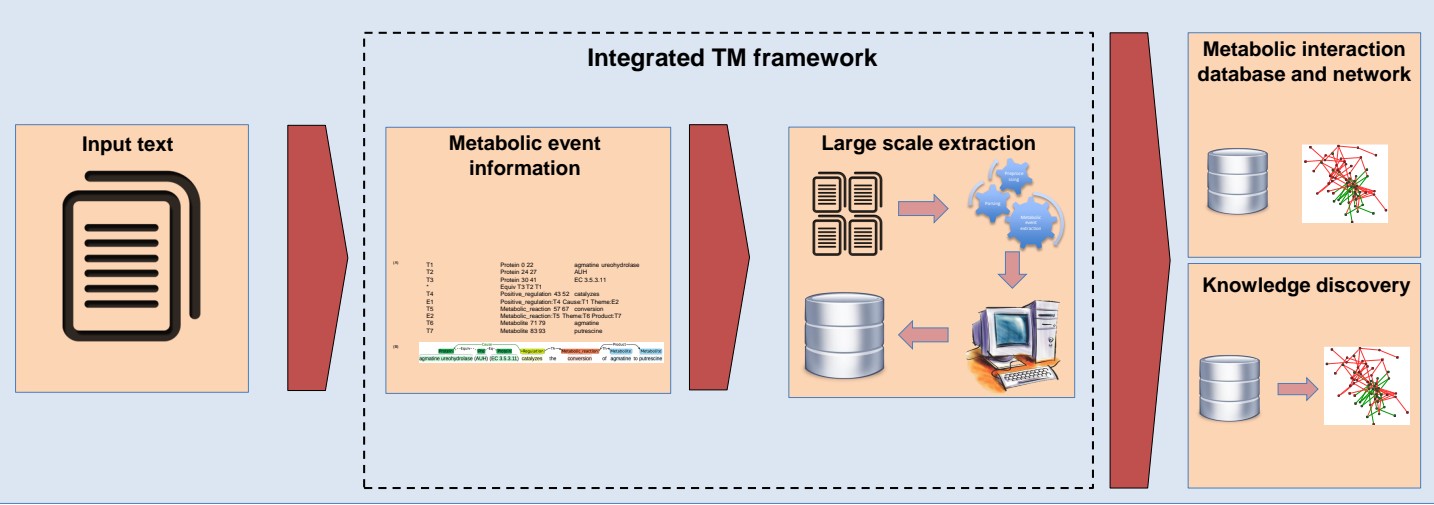

**Figure 6** The integrated TM framework application.

ME corpus, F-scores of 59.15% and 48.59% were obtained for metabolic production and consumption, respectively. This indicates practical performance of the MEE module in the detection of metabolic entities. In the comparative study of corpus sizes, the MEE module showed a high F-score and a high recall when the size increased. The correlation of its overall performance implied the extensibility of the integrated TM framework which could be achieved by increasing the size of the corpus.

With regards to the entity tagger for GP and metabolites on the test corpus, the overall performance showed F-score for more than 80% as illustrated for the Superpathway of leucine, valine, and isoleucine biosynthesis. Concerning on the evaluated MINR module on the test corpus, F-score of above 70% was achieved in mapping of enzyme–metabolite interactions through network reconstruction. Furthermore, the integrated TM framework could be used to successfully reconstruct the Superpathway of leucine, valine, and isoleucine biosynthesis with comparable performance to another TM system in terms of enzymes, metabolites, and enzyme–metabolite interactions. Finally, when the integrated TM framework was applied with an EcoCyc extraction for reconstructing a metabolic interaction network, reasonable precisions of enzyme (69.93%), metabolite (70.63%) and enzyme–metabolite interaction (46.71%) were obtained. We believe that this study can be the first such TM framework for developing further automation tools for assisting in metabolic network reconstruction. In addition, this TM framework is beneficial for general usability because it can run on Linux, Window and MAC systems. Based on our investigations, finer details of metabolic interaction, such as types of interactions, locations, pathways and species could be deduced from metabolic events and linguistic patterns. This rich information would allow us to build a more accurate representation of metabolic interactions and more sophisticated metabolic network reconstruction. For a future plan, other databases, e.g., MetaCyc, contain thousands of different organisms which can be used for evaluation and application of an integrated TM framework for a species-specific metabolic interaction network. Full text of paper is recommended for future development

of corpus. Due to the limitation of the current tool (e.g., BANNER) used in this study, the NLP-based approach should be further investigated and implemented for increasing the overall performance of the integrated TM framework. An API for implementation of user-defined algorithms will also be provided, including a user interface and a web-service for event annotation.

**Abbreviations**

| | |
|---|---|
| **TM** | Text Mining |
| **PN** | Pyridoxine |
| **PL** | Pyridoxal |
| **HTP** | 4-hydroxy-l-threoninephosphate |
| **TEES** | Turku Event Extraction System |
| **GP** | Gene and Protein |
| **SD** | Standard deviation |
| **GREC** | Gene Regulation Event Corpus |
| **BioNLP-ST** | BioNLP Shared Task |
| **NLP** | Natural Language Processing |
| **ME** | Metabolic Entities |
| **MEE module** | Metabolic Event Extraction module |
| **MINR module** | Metabolic Interaction Network Reconstruction module |

## ACKNOWLEDGEMENTS

We would like to thank Mr. Yutthanattee Tohreh for assisting in the manual annotation process and Mr. Sean Kortschot for proofreading the draft version of the manuscript. We also would like to thank Mr. David H. Cook for proofreading the revised version of manuscript.

### Funding

Financial support was provided by Soochow University (grant no. Q410700111), the National Natural Science Foundation of China (NSFC) (grant no. 31200989), King Mongkut's University of Technology Thonburi (KMUTT), Prepproposal Research Fund (grant no. PRF 4/2558), Faculty of Science, Kasetsart University, and the Thailand Research Fund (grant no. TRG5880245). The funders had no role in study design, data collection and analysis, decision to publish, or preparation of the manuscript.

### Grant Disclosures

The following grant information was disclosed by the authors:
Soochow University: Q410700111.
National Natural Science Foundation of China (NSFC): 31200989.
King Mongkut's University of Technology Thonburi (KMUTT).
Prepproposal Research Fund: PRF 4/2558.
Faculty of Science, Kasetsart University.
The Thailand Research Fund: TRG5880245.

## Competing Interests

The authors declare there are no competing interests.

## Author Contributions

- Preecha Patumcharoenpol performed the experiments, analyzed the data, wrote the paper, prepared figures and/or tables, reviewed drafts of the paper.
- Narumol Doungpan and Asawin Meechai analyzed the data, reviewed drafts of the paper.
- Bairong Shen contributed reagents/materials/analysis tools, reviewed drafts of the paper.
- Jonathan H. Chan conceived and designed the experiments, reviewed drafts of the paper.
- Wanwipa Vongsangnak conceived and designed the experiments, contributed reagents/materials/analysis tools, wrote the paper, reviewed drafts of the paper.

## Data Availability

Supplement: an integrated text mining framework for metabolic interaction network reconstruction: www.sbi.kmutt.ac.th/~preecha/metrecon.

## Supplemental Information

Supplemental information for this article can be found online at http://dx.doi.org/10.7717/peerj.1811#supplemental-information.

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
