# Peer review of "An integrated text mining framework for metabolic interaction network reconstruction"

_PeerJ, doi:10.7717/peerj.1811_

## Round 0.1 · original submission · Major Revisions

As you can see, both reviewers found that your manuscript is addressing an important problem in the field, but that each had concerns about the study as presented. In general, these comments cite a lack of a clear description of some of the methods used, justification for choice of particular parameters and construction of experiments, and comparison with other current state-of-the-art methods. This makes it difficult to judge completely the validity of the overall claims of this study.

Reviewer 1 ·

Basic reporting

This paper presents a fully realized approach for constructing metabolic networks from text. The effort includes annotating a new corpus for relevant physical entities and events, adapting and retraining machine-learning methods for the task, integrating their results, and applying the system at scale. The created resources are of potential value to the community, the technical approach is appropriate, and the evaluation supports the value of the overall approach for supporting pathway curation efforts. I believe this manuscript will make a valuable addition to the domain literature.

I have some concerns regarding the positioning of the study with respect to previous work and specific claims of novelty, and a number of minor points require further clarification.

Regarding novelty, the authors use the event extraction framework and related tools (esp. TEES) yet write "there have been limited studies to date on extracting metabolic pathways [...] from free texts" and omit discussion of the 2013 pathway curation event extraction task (cited in passing as [18]) which includes metabolic reactions and to which TEES has been previously applied. The relationship of the introduced approach with this closely related prior work should be discussed and specific points of novelty identified in more detail. Similarly, the differences to the related systems discussed in background should be made more explicit. (Is the primary difference the use of ML components?)

Regarding position with respect to previous findings, the presented entity and event extraction performance numbers are almost exclusively for the proposed approach, and there is only limited discussion of how these compare to previous work. Although results at different tasks are never fully comparable, mention of state-of-the-art results at closely related tasks (gene/protein name recognition, metabolic event extraction) should be presented for reference and discussed.

The level of English is adequate, but the text would benefit from further editing, including a spell-check (e.g. "sophisicated") and particular attention on the use of singular and plural forms.

Experimental design

The experimental design is generally apt, but some points should be further clarified to facilitate replication:

- Please provide detailed definitions of the annotated physical entity and event types, preferably with respect to relevant domain ontologies (e.g. Gene Ontology, ChEBI)
- Please describe how these definitions relate to those in closely related resources (e.g. gene/protein corpora such as GENETAG, event corpora)
- What (hyper)parameter values were used for the machine learning methods (e.g. CRF regularisation parameter, TEES parameters)?
- How were these parameters selected?
- In cross-validation experiments, at what granularity were examples split into train/test (e.g. sentence, document)?
- How was TEES "retrofitted" to the task? What modifications other than retraining were performed?
- Which evaluation tools were used, and where are these available?
- When evaluating the event extraction tool, were predicted or manually annotated physical entity mentions used as input?

There is a surprising asymmetry in the experiments: event extraction is evaluated using cross-validation (Table 4) but entity tagging and network construction using separate train/test corpora (Tables 5 and 6). Please detail the reasons for this experimental setup (or consider applying the same setup to all components).

Validity of the findings

The authors are to be commended for providing all data and code for their study. However, when making your data and source code available, please identify also the appropriate open (source / data) licenses that apply to each (see https://peerj.com/about/policies-and-procedures/#data-materials-sharing).

Additional comments

Miscellaneous minor comments:

- [28]: "biological entities (e.g., genes, proteins, biomolecular pathways and network relationships)": pathways and relationships as examples of are entities is somewhat unusual in this context. You may wish to restate this.
- [82-83]: "the available tools have various restrictions, such as not being freely available or working only with predefined data": this appears to be contradicted by TEES, which you apply. Please clarify.
- [172-173]: "we selected only enzymes entities out of all possible GP entities throughout metabolic events.": how was this selection performed? How reliable is it?
- [199-200]: "unique enzyme-metabolite interaction pairs were recursively merged together and iteratively mapped onto a metabolic interaction network": unclear, please restate (what is "recursively" here?).
- [288-290]: "it is possible to deduce the types of enzymatic reactions (e.g., phosphorylation, deamination, and transamination) using name of substrate and product.": please clarify if this is meant as a general claim and provide examples.
- [384]: "we did deploy a normalization method": missing "not"
- [Figure 5]: The results presented here appear substantially lower than those discussed in the rest of the paper. Please provide discussion and explanation of this difference.

Reviewer 2 ·

Basic reporting

The article conforms PeerJ rules.

Experimental design

This paper tackles a difficult and important question, which is how information extraction from scientific papers could contribute in the reconstruction of metabolic interaction networks.
The approach consists in the integration of state of the art methods, BANNER for the recognition of biological entities and TEES for the extraction of the biological event in the text. The interaction network is obtained by merging the individual events.
A manually annotated corpus has been designed as reference corpus for the training and the evaluation of the event extraction method. External resources (EcoCyc and papers) have been used for the evaluation of network reconstruction. The evaluation measures are based on standard measures in the domain. Many experiments described in the paper and in the supplementary material illustrate the quality of the approach.

However the paper suffers from lack of precision on some scientific issues and lack of justification of some choices.
More precisely, the design of a new corpus with a new annotation schemata raise questions. The paper should better justify the reason why the existing corpora such as "Pathway Curation" (PC) (BioNLP-ST'13) or a subpart of it is not useable here. What kind of new knowledge becomes predictable with the authors corpus that would be not predictable with the other corpora? The author corpus consists of abstracts without the titles. The reason for not using titles and full papers should be explained. Is the useful information for metabolic network reconstruction fully described in the abstracts? The size of the corpus in number of events is quite small compared to the standards in IE tasks (e.g. half of the PC corpus event number). It should be justified by the regularity of the event descriptions that may be easier to extract.
The paper details the evaluation of the author event extraction method on their corpus. It would be useful to compare these results with the state of the art. One way to would be the application of the method to the Shared Task corpus (e.g PC) for which method evaulations have been published and then the comparison of its results with the results of the other methods applied to the same corpus.
The corpus annotation schemata is simple (four types of events). The paper should explain how it has been designed and if it is sufficient to capture the relevant biological information and distinguish among close events. It should also justify why the standard and richer representations of Gene Ontology and System Biology Ontology were not appropriate here.
In particular, SBO details the biochemical reactions (conversion, removal or transfer of a chemical group) that look relevant to the paper objectives. The events called production, consumption, reaction and regulation in the paper maybe not enough detailed to disambiguate among the candidate extracted reactions.
The definition of the event types is shortly given in the on-line guidelines file, it would be useful that the paper also gives them and clarifies the kind of biological event that the four types represent, in particular regulation. The regulation type of event is used here for annotating processes like catalysis (example of figure 2), which seems to be misleading from a biological point of view. Catalysis may be regulated by an entity but considering itself as a regulation is surprising.
It appeared by randomly selecting some abstract of the annotated reference corpus that it contains errors that should be revised before publication. For instance, the abstract of paper PMID-177406 describes the well-known transfer of a phosphate from phosphoenolpyruvate to glucose by the PTS system (glucose transporter + transfer). This event is annotated as a metabolic reaction between the phosphotransferase of the PTS and the glucose. The role of the phosphoenolpyruvate as phophate donnor should be annotated.
The measure of the agreement inter-annotator is provided by the paper. Its level (70-90) is rather high but it has been computed from a small subset (20 documents). It could explain the high variability of the agreement levels among entities and events.
The choice of BANNER and TEES and information extraction methods should be motivated.
The way the methods are evaluated against EcoCyc information is too vague (243 - 247 and 347-390). The type of the EcoCyc information and the [45] paper information that is used should be explained. The objects that are compared should be better defined.

Validity of the findings

The validity of the findings in the event extraction and in the metabolic network reconstruction is hard to evaluate du to the lack of clarity of the experimental settings. The approach looks reasonable but should be more detailed.

Additional comments

Detailed comments
* * *
The paper would be easier to read if the two methods (module 1 and module 2) and the corpus would be named. The abstract could be less technical. It could first describe the context of the work and not be too detailed about the experimental framework (e.g. five cross validation).
In section "Construction of metabolic entity corpus" EcoCyc is presented as the source of information for the method before it is introduced.
The schemata of event hierarchy and biological entity hierarchy could be provided in the paper in order to clarify their definitions (metabolite, enzyme, GP). For instance is an enzyme a GP?

---

## Round 0.2 · Minor Revisions

As indicated by the reviewers, you have successfully satisfied the major concerns that they had with the original submissions. Both reviewers have a suggested list of minor revisions that primarily have to do with clarity and proper English language usage that should be straightforward to satisfy.

Reviewer 1 ·

Basic reporting

The submission adheres to PeerJ policies.

Experimental design

No comments.

Validity of the findings

No comments.

Additional comments

(From my review of the initial submission, and still valid) This paper presents a fully realized approach for constructing metabolic networks from text. The effort includes annotating a new corpus for relevant physical entities and events, adapting and retraining machine-learning methods for the task, integrating their results, and applying the system at scale. The created resources are of potential value to the community, the technical approach is appropriate, and the evaluation supports the value of the overall approach for supporting pathway curation efforts. I believe this manuscript will make a valuable addition to the domain literature.

In this revision, the authors have carefully addressed my main concerns with the initial submission, in particular clarifying the position of their study with respect to previous work, as well as addressing all but few of the minor issues that I raised.

Two questions (11. and 15.) received an answer in the authors' response letter but not in the manuscript (as far as I can tell):

11. How was TEES "retrofitted" to the task? What modifications other than retraining were performed?

Ans: Regarding TEES used for metabolic event extraction, several parameters in TEES e.g., entity category (Gene type of entities) were fixed and these were regarded as predefined data. To solve, we needed to retrofit TEES by introducing other types of entities (e.g., metabolites) and allowing to detect metabolic events in virtual machine image provided at http:// www.sbi.kmutt.ac.th/~preecha/metrecon/. 


I could not find these details (e.g. "retrofit ... by introducing other types of entities ... and allowing to detect metabolic events", i.e. if I understand correctly, "retrofitting" involves reconfiguration of the entity and event types plus retraining) in the manuscript. Please add this information.

15. The authors are to be commended for providing all data and code for their study. However, when making your data and source code available, please identify also the appropriate open (source / data) licenses that apply to each (see https://peerj.com/about/policies-and-procedures/ #data-materials-sharing).

Ans: Thanks for your comments and suggestion. We already added the Creative Commons Attribution-ShareAlike 4.0 International License for the metabolic entities corpus and the Apache License, version 2.0 for source code at www.sbi.kmutt.ac.th/~preecha/metrecon.

The specific licenses should be identified in the manuscript (not just on the website).

I have a few minor issues with the revised manuscript, detailed in the following:

- "This study presents the first effort to reconstruct a metabolic interaction network based on an integrated TM framework."

This appears to overstate the novelty wrt. what is presented elsewhere in the manuscript and should be further qualified (e.g. by adding "open source" or similar), as previous closed or non-available systems discussed in the manuscript have addressed (effectively) the same task.

- "Directing closely related prior works"

Unclear, please rephrase "directing".

- "diverged recall and precision"

Please clarify "diverged".

- "PathText [22] is not open access"

The concept of Open Access applies poorly to software. Perhaps "open source" or just "openly available".

- "... presented in Table 2 ... Table 2 presents ..."

Please remove the redundancy.

- "For the MEE module, we assessed ... ME corpus size", "MEE module evaluation on ME corpus size", and similar

Please rephrase for clarity. You may wish to emphasize the phrase "learning curve", which is more readily understood than "evaluation on corpus size".

- "a comparison of PC task - BioNLP-ST’13 [18] and our study was performed for performance evaluation as in example event of positive regulation. Consequently, the achieved F-scores were similar with values of 39.23 and 36.69, respectively."

This is an interesting additional experiment, but it is difficult to understand exactly what was done from this brief description. Please add further detail identifying the specific experimental setup.

Despite improvements to the English, many minor language issues remain, most frequently relating to the use of determiners and pluralization. The following is an incomplete list of issues from the first few pages:

- Evaluation of MEE module -> Evaluation of the MEE module
- Mapping of ... interaction -> Mapping of ... interactions
- performance of MINR module -> performance of the MINR module
- extract the relationship between ... entities -> extract the relationships between ... entities
- genedisease -> gene-disease
- performing simple interaction extractions -> performing simple interaction extraction
- pathways and regulatory network -> pathways and regulatory networks
- Focusing on metabolic relationship, Pathway Curation (PC) Task -> Focusing on metabolic relationships, the Pathway Curation (PC) Task
- through curation of events -> through the curation of events

These language issues are not a serious hindrance to understanding the manuscript, and I do not believe that perfect English should be a prerequisite to publication. Nevertheless, to improve the perceived overall quality of their paper, I recommend the authors perform a further careful round of editing with a native speaker.

Reviewer 2 ·

Basic reporting

The authors have carefully revised their manuscript. They have adequately adressed all issues that I raised with the original version of their manuscript.
Minor corrections should be addressed before publication, see below.

Experimental design

No comments

Validity of the findings

No comments

Additional comments

Minor comments
line 78: "its corpus" is ambiguous.
line 82: "For other generic system, e.g. the GeneWay system is used" should be rephrased.
line 96: "predefined data" is unclear.
line 109, 335: abovementionned -> above mentionned
line 133: A ME -> The ME
lines 136, 166, 169 : calling the annnotators A and B is not needed since there is no reference to one independently of the other. The "two annotators" is sufficient here.
line 167: guideline -> guidelines
line 172: agreement between annotator A and B -> inter-annotator agreement
lines 176-177 are unclear, whose interpetation is it about?
line 206: "we retrofitted" TEES from its original version" is unclear.
lines 210-213: the choice of TEES should be moved at the beginning of the paragraph.
line 224: the redundancy of the extracted events across papers could be mentioned to clarifiy the reason why the nodes are rearranged and deleted.
line 247: (iii) what does "some matching" means? How are partial matchings of the arguments penalized?
line 299: A constructed ME corpus -> "the ME corpus", or A manually annotated / reference corpus called ME, the ME reference corpus ...
line 338: "as in example event of positive regulation" should be rephrased.
lines 352-353: "regularity is introduced for..." is unclear.
lines 433-434: From the overall performances ... they performed well" should be rephrased.
line 461: what does "the feasible integrating NLP-based approcach mean?

---

## Round 0.3 · accepted · Accept

Thank you for the careful attention to improving the clarity of the manuscript.